# A Systematic Review and Meta-Analysis of the Incidence and Risk Factors for Major Adverse Cardiovascular Events in Patients with Unrepaired Abdominal Aortic Aneurysms

**DOI:** 10.3390/biomedicines11041178

**Published:** 2023-04-14

**Authors:** Chinmay Sharma, Tejas P. Singh, Shivshankar Thanigaimani, Domenico Nastasi, Jonathan Golledge

**Affiliations:** 1Queensland Research Centre for Peripheral Vascular Disease, College of Medicine and Dentistry, James Cook University, Townsville, QLD 4811, Australia; chinmay.sharma@my.jcu.edu.au (C.S.); shiv.thanigaimani@jcu.edu.au (S.T.); 2Department of Vascular and Endovascular Surgery, Townsville University Hospital, Townsville, QLD 4811, Australia; tejas.singh@my.jcu.edu.au; 3The Australian Institute of Tropical Health and Medicine, James Cook University, Townsville, QLD 4811, Australia; 4Department of Vascular and Endovascular Surgery, Gold Coast University Hospital, Southport, QLD 4215, Australia; domenico.nastasi@my.jcu.edu.au

**Keywords:** abdominal aortic aneurysm, cardiovascular death, myocardial infarction, stroke

## Abstract

Major adverse cardiovascular events (MACE), including myocardial infarction (MI), stroke and cardiovascular death, cause substantial morbidity and mortality. This review assessed the incidence rate of MACE and the association with modifiable risk factors (diabetes, hypertension) and medication use (aspirin, statins) in patients with unrepaired abdominal aortic aneurysm (AAA). Electronic databases were searched systematically for observational studies reporting the incidence of MI, stroke or cardiovascular death in patients with unrepaired AAAs. The primary outcome was cardiovascular death reported as an incidence rate (events per 100 person-years (PY)). Fourteen studies, including 69,579 participants with a mean follow-up time of 5.4 years, were included. Meta-analysis revealed the overall incidence of cardiovascular death, MI and stroke of 2.31 per 100 PY (95% CI, 1.63–3.26; I^2^ = 98%), 1.65 per 100 PY (95% CI, 1.01–2.69, I^2^ = 88%) and 0.89 per 100 PY (95% CI, 0.53–1.48, I^2^ = 87.0%), respectively. The mean rates of statin and aspirin prescriptions were 58.1% and 53.5%, respectively. In conclusion, there is a substantial incidence of MACE in patients with unrepaired AAA, but the prescription of preventative medication is suboptimal. Greater emphasis should be placed on secondary prevention in this population.

## 1. Introduction

Abdominal aortic aneurysm (AAA) causes approximately 200,000 deaths per year worldwide due to aortic rupture or complications of surgical repair [1]. Surgical repair is the only established treatment for AAA [1]. Current guidelines recommend elective repair of asymptomatic AAAs measuring ≥50 mm in women and ≥55 mm in men [2]. Many countries, including the UK, USA, Sweden and other European countries, have screening programs to identify AAA [3]. These screening programs mainly identify small AAAs below thresholds for repair that enter imaging surveillance [4]. A recent meta-analysis of randomised controlled trials suggested that screening reduces AAA-related mortality but not all-cause mortality [4]. AAA screening programs could potentially be more valuable if they could also prevent non-aneurysmal causes of death.

Coronary heart and cerebrovascular diseases are frequent causes of morbidity and death in patients with small AAA [5]. A previous systematic review estimated that the incidence of cardiovascular death in patients with a small AAA was approximately 3% per year [5]. There has, however, been no recent systematic assessment of the overall risk of major adverse cardiovascular events (MACE), including myocardial infarction (MI), stroke and cardiovascular death in patients with unrepaired AAA. A recent prospective study of Australian patients with an aortic diameter of 30–54 mm reported an incidence of MACE of 38% over five years [6]. Whether these findings are unique to this population is unclear. The risk factors for MACE in people with AAA have also not been systematically reviewed. Understanding this could be valuable to revising aneurysm screening programs. A prior study suggested that patients with unrepaired AAA may have unique risk factors for MACE not identified in those without AAA [7]. Most AAAs contain intra-luminal thrombus, and it has been previously reported that patients with large volumes of thrombus have the greatest risk of MACE [7]. Due to their unique cardiovascular profile, a systematic assessment of the risk factors for MACE in patients with unrepaired AAA is needed. 

This systematic review aimed to provide an up-to-date synthesis of the incidence and risk factors for cardiovascular death, MI and stroke in patients with unrepaired AAA. 

## 2. Methods

### 2.1. Search Strategy

This systematic review was performed according to the Preferred Reporting Items for Systematic Review and Meta-Analyses (PRISMA) statement [8]. The study protocol was registered with PROSPERO (Registration ID—CRD42022320543). The literature search strategy was developed by one author (CS). The databases Medline and Scopus were searched on 18 April 2022. The search strategy used is shown in the Appendix A. No date or language restrictions were applied. To be eligible for inclusion, studies needed to report the incidence of either MI, stroke or cardiovascular death in patients with unrepaired AAAs and have an observational design. Studies that focused on outcomes following AAA repair were excluded. All other studies, including those that reported outcomes for unrepaired and repaired AAAs, providing that data was available for patients with unrepaired AAA independently, were eligible for inclusion. Studies that did not include patients with AAAs, and case studies, letters and those studies that did not report MACE were excluded. Included articles were identified by two authors (CS and TS). 

### 2.2. Data Extraction and Outcomes

Two authors (CS and TS) extracted data on a customised spreadsheet. The primary outcome of the study was the incidence of cardiovascular death, defined as death caused by MI, stroke or other cardiovascular causes (excluding AAA-related mortality where specified). Secondary outcomes were MACE, MI, stroke and AAA-related mortality. MACE was defined as non-fatal MI, stroke or cardiovascular death. AAA-related mortality was defined as death due to AAA rupture or within 30 days of AAA repair. The following data were also collected from the included studies: Study design, sample size, age, sex, duration of follow-up, study outcomes, diabetes, hypertension, smoking history, current smoking status, AAA diameter, systolic and diastolic blood pressure, low density lipoprotein cholesterol, prescription of statins and aspirin and AAA-related deaths.

### 2.3. Quality Assessment

Two authors (CS and TS) independently assessed the quality of the included studies using a quality assessment tool adapted and modified from a previously published systematic review (Appendix A) [9]. This assessed study objective, study design, sample size estimation, reporting of unrepaired AAA, reporting of MACE, primary outcome, participant characteristics, follow up and statistical methods. Any inconsistencies were resolved through discussion between the authors until a consensus was reached. The scores were specified based on the predetermined questionnaire, with responses scored as Yes = 2, Partial = 1 and No = 0. The final quality assessment scores were expressed in terms of percentage (%). Percentage scores of <50%, 50–70% and >70% were considered high, moderate and low risk of bias, respectively. 

### 2.4. Data Analysis

A minimum of five studies reporting an outcome were required to be eligible for meta-analysis. We anticipated heterogeneity between studies, and therefore random-effects models were used. The model was developed using Mantel-Haenszel’s method, which adjusts for confounding factors. The results of individual studies were combined by weighting each study by its inverse variance to reflect the amount of uncertainty in the individual studies. Data were expressed as incidence per 100 person years (PY) with 95% confidence intervals (CI). The I^2^ index was used to assess the degree of statistical heterogeneity between studies, with I^2^ > 50% accepted to denote high heterogeneity. Meta-analyses were conducted using the ‘meta’ and ‘dmetaR’ packages of R software version 4.0.3, R Foundation for Statistical Computing, Vienna, Austria. Leave one out analysis was performed by removing one study at a time to determine the incidence per 100 PY. The influence of any single study was represented by Cook’ distance of greater than 0.50 to be considered influential. A *p*-value of <0.05 was considered statistically significant.

## 3. Results

### 3.1. Characteristics of Studies and Included Participants

A total of 14 studies included 69,579 participants with unrepaired AAA with a mean follow-up time of 5.4 years (Figure 1). Nine studies were prospective cohort studies [6,7,10,11,12,13,14,15,16], and five studies were retrospective cohort studies [17,18,19,20,21]. Twelve studies reported the incidence of cardiovascular death in 35,936 participants [6,7,10,11,12,13,14,15,16,17,20,21]. Five studies reported the incidence of non-fatal MI and non-fatal stroke in 35,860 participants [6,7,10,11,12,13,14,15,16,17,20,21]. Where reported, a weighted average of 41.1%, 11.1%, 75.7%, and 27.4% of the participants had hypertension, diabetes, had ever smoked, and currently smoked, respectively (Table 1). A mean of 53.5% and 58.1% were prescribed aspirin and a statin, respectively (Table 1). It is worth noting that the weighted average percentage of males in the population included was 81%. Separate cohort-specific data was not available for the smaller proportion (19%) of female participants. There was an under-representation of Asian, African and South American populations.

### 3.2. Risk of Bias of Included Studies

As reported in Table 2, five studies were deemed to be at low risk of bias [6,7,11,13,15], eight studies at moderate risk of bias [10,12,14,16,17,18,19,21], and one at high risk of bias [20]. All included studies reported the objectives clearly and described the study design. None of the included studies reported sample size estimates. Nine studies reported the repair status of AAAs as unrepaired [6,7,10,11,12,13,15,16,21]. Six studies reported the incidence of all three outcomes of MI, stroke and cardiovascular death [6,7,13,14,16,21]. Thirteen studies reported the primary outcome of cardiovascular death, eleven of which noted this was adjudicated by an independent assessor [6,7,10,11,12,13,14,15,16,17,19,20,21]. Eight studies reported all participant characteristics [6,7,11,14,15,17,18,19]. Five studies did not report the characteristics of the subpopulation of patients with unrepaired AAA [12,13,16,20,21]. In these cases, the corresponding authors were contacted for the relevant information, but none replied. Twelve studies reported the length of follow-up [6,7,10,11,12,13,14,15,16,17,19,20,21]. Twelve studies used statistical methods to control for confounding [6,7,10,12,13,14,15,17,18,19,20,21]. Overall, the assessment of the included studies indicated a moderate mean quality score of 65.1 ± 13.7%. 

### 3.3. Incidence of Cardiovascular Death, MI and Stroke

The number of cardiovascular deaths reported in each study is reported in Table 3. Ten studies determined cardiovascular death by accessing medical record databases [6,7,10,11,12,13,14,15,17,20]. In two studies, the cause of death was determined by post-mortem examination [16,21]. Quantitative analysis revealed an overall incidence of cardiovascular death of 2.31 per 100 PY (95% CI, 1.63–3.26; I^2^ = 98%) (Figure 2). The incidence of MI and stroke were 1.65 per 100 PY (95% CI, 1.01–2.69, I^2^ = 88%) and 0.89 per 100 PY (95% CI, 0.53–1.48, I^2^ = 87.0%), respectively (Figure 2).

The number of AAA-related deaths in each study is reported in Table 3. Quantitative analysis revealed an overall incidence of AAA-related death of 0.49 per 100 PY (95% CI, 0.21–1.14, I^2^ = 98.0%) (Figure 2).

A leave one out analysis was performed to assess whether the incidence of cardiovascular death was influenced by any single study. The results are presented in Table 4. Leave one out analysis showed that one study by Oliver-Williams et al. had the most influence on the primary outcome. There was a substantial change in the incidence of cardiovascular death when this study was omitted (Table 4). 

Furthermore, Brown et al. and Brady et al. reported on subsets of patients from the same cohort; hence, sensitivity analysis was important to determine their influence. It was seen that neither study had a significant impact on the total incidence of cardiovascular death. 

### 3.4. Risk Factors for Events

Two studies examined risk factors for events within a cohort of AAA patients. Takigawa et al. found that participants with diabetes had an increased risk of MI (OR = 3.5, 95% CI 1.1–19.3, *p* = 0.048) [19]. Hypertension and obesity were not significantly associated with the risk of MI, stroke or cardiovascular death [19]. Parr et al. reported that participants with a large aneurysm thrombus volume (≥25.0 cm^3^) have an increased risk of MACE (RR = 2.3, 95% CI 1.01–5.24, *p* < 0.05) [7]. 

Meta-regression demonstrated no statistically significant association between the incidence of cardiovascular death and diabetes (Intercept = −3.61, SE = 0.01, *p* = 0.35), hypertension (Intercept = −3.46, SE = 0.004, *p* = 0.27) or prescription of aspirin (Intercept = −4.83, SE = 0.02, *p* = 0.77) or statins (Intercept = −5.74, SE = 0.05, *p* = 0.62) (Table 5). 

There was, however, a statistically significant reduction in the incidence of cardiovascular death when correlated with the year of publication (Intercept = 95.49, SE = 0.02, *p* = 0.01). As one would expect, the incidence of cardiovascular deaths decreased over time (Table 5). Myocardial infarction was significantly more common in studies with lower aspirin prescription (Intercept = 6.27, SE = 0.01, *p* < 0.01). None of the assessed risk factors were associated with occurrence of either myocardial infarction or stroke alone.

## 4. Discussion

This review suggests that the mean incidence of cardiovascular death, MI and stroke in patients with unrepaired AAA are 2.31 (95% CI 1.63–3.26), 1.65 (95% CI 1.01–2.69) and 0.89 (95% CI 0.53–1.48) per 100 PY respectively. The mean incidence of cardiovascular mortality in the general global population was reported to be 0.2 per 100 PY in developed countries in 2017 [23]. This implies that the rate of cardiovascular death is 12 times greater in patients with unrepaired AAA than in the general population. This highlights the need to focus on secondary cardiovascular prevention in any person diagnosed with a small AAA. The reported weighted average rates of statin (58%) and anti-platelet medication (54%) prescription and current smoking (27%) indicate much more needs to be done to optimise secondary prevention in the population with unrepaired AAA. The range of treatments effective in reducing the risk of MACE is rapidly expanding and now includes low-dose rivaroxaban, new low-density lipoprotein cholesterol reducing drugs, anti-inflammatory medications and multiple novel drugs to treat diabetes [24]. It is vital that programs that are effective in implementing cardiovascular prevention are developed.

A number of included studies identified risk factors for cardiovascular events in patients with unrepaired aneurysms, including obesity, larger AAA diameter and aneurysm thrombus [7,12,17]. A more complete understanding of the risk predictors for MACE could enable risk models to be established relevant to patients with small AAA, which could contribute to better targeting secondary prevention treatments. The meta-regression found a significant association between reduced cardiovascular death and year of publication. A significant association was also found between increased myocardial infarction and lower aspirin use. There were no significant associations between diabetes, hypertension, prescription of statins and MACE (Table 5). This finding is likely confounded by small numbers and the lack of information on the effectiveness of the drugs in controlling modifiable risk factors, drug adherence and dose.

Metaregression showed a significant correlation between cardiovascular death and the year of publication, showing that with time, there has been a significant downtrend in the incidence of cardiovascular death in patients with AAA. This suggests that secondary prevention through medical management of patient risk factors improved over time, leading to a reduction in cardiovascular deaths.

As shown by the sensitivity analysis, Oliver-Williams et al. strongly influenced the overall incidence rate of cardiovascular death, owing to the large population size. This study was conducted as a population screening program in otherwise healthy, asymptomatic men, which may explain the relatively lower incidence rate of cardiovascular death in this population. There was also a moderate risk of bias in this study, which may have played a role in the skewed results. 

Of all included studies, Brady et al. showed the highest incidence rate of cardiovascular death (4.47 per 100 PY). Still, due to a small sample size, it had a relatively small influence on the total incidence rate. Omitting the study by Brown et al. had almost no influence on the incidence of cardiovascular death. Despite having overlapping patient cohorts, neither study had a substantial influence on the final results. 

There was significant variation in the reported incidence of cardiovascular death, MI and stroke in the included studies. This variation was likely reflective of varying definitions for MI and stroke. Some studies reported MI and stroke as causes of mortality, and others reported non-fatal outcomes. Variation in follow-up length was also an important reason for the variation in reported outcomes, which was accounted for by comparing incidence rates in events per 100 PY. Other factors contributing to heterogeneity included varying participant age, population identification (screening vs. hospital referral) and the different prevalence of cardiovascular risk factors. 

Antiplatelet (aspirin), lipid-lowering (statins), and antihypertensive medications are the mainstay of preventing cardiovascular events. The Meta-regression found that higher MI rate was significantly associated with lower aspirin prescription. Both aspirin and statin therapy have been independently validated as effective in preventing cardiovascular events. Evidence supports their use and points to their under-prescription in patients with AAA [25,26,27]. The weighted average percentage of included patients prescribed aspirin (53.5%) and statins (58.1%) is comparable to data published in previous studies. Lloyd et al. found that an antiplatelet medication was indicated in 92% of AAA patients, but only 60% of the patients were prescribed this medication [28]. Similarly, they found that statins were indicated in 75% of AAA patients, but only 41% of the patients were prescribed these drugs [28]. The disparity in indication and prescription rates demonstrates the suboptimal secondary prevention of MACE in AAA patients. Due to the lack of studies assessing the direct potential benefit of cardiovascular risk-lowering medications in AAA patients, no tailored guidelines exist on cardiovascular risk prevention in AAA patients.

The most effective way to integrate secondary cardiovascular prevention into an AAA screening program is currently unclear. Suggested approaches include integration with primary practice services or hospital-based cardiovascular specialists, stand-alone telehealth programs or incorporation within the screening visits. Further research is needed to clarify which approach participants and clinicians favour. 

The results of this systematic review were limited by several factors. Firstly, the number of studies included (14), and the number of included patients (69,579) was small. Furthermore, after the three studies with the largest patient cohorts were removed, the cumulative number of included patients was only 6712 patients. This highlights that the results of this review are dominated by the populations reported by Nicolajsen et al., Ko et al. and Oliver-Williams et al. and thus may not be generalisable to other populations. The participants were, on average, 81% men, and the findings may not be relatable to women with unrepaired AAA. Finally, nationality in this review was reported as predominantly British, Danish and Australian. A greater diversity of patient cohorts would improve the generalisability of these results. 

In conclusion, this review suggests that the incidence of MACE is substantial in patients with unrepaired AAA. More effective methods are needed to implement secondary cardiovascular prevention in this population. Metaregression did not find a significant association of cardiovascular death with the assessed risk factors, hypertension, diabetes, aspirin and statin use. This was likely confounded by the small sample size and lack of drug adherence and dose information. At present, the lack of consensus regarding guidelines on the long-term medical management of cardiovascular risk factors for these patients is a key issue to address. Given the high risk of MACE in AAA patients and the suboptimal implementation of prevention strategies, greater emphasis should be placed on mitigating this risk through lifestyle changes and optimising prescription rates of risk factor-lowering medications.

## Figures and Tables

**Figure 1 biomedicines-11-01178-f001:**
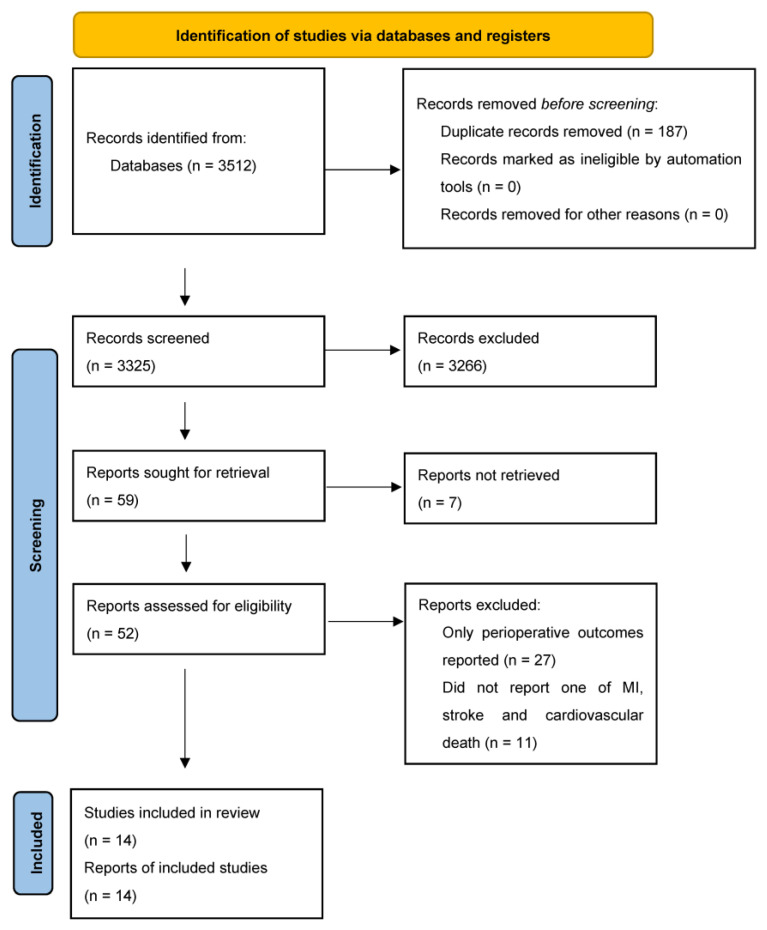
PRISMA Flow Diagram.

**Figure 2 biomedicines-11-01178-f002:**
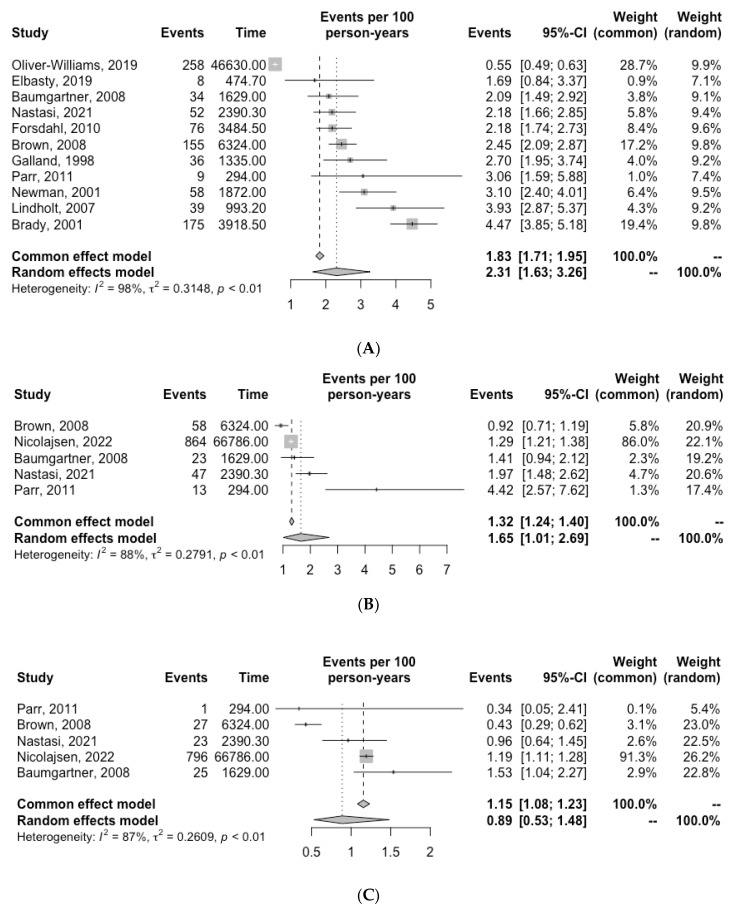
Incidence of (**A**) cardiovascular death, (**B**) myocardial infarction (**C**) stroke and (**D**) AAA-related death for patients with unrepaired AAA. Grey squares indicate the incidence of events per 100 person-years (PY) and horizontal lines indicate the 95% CI for each study. The diamond represents the respective incidence for patients with unrepaired AAA calculated with random-effects meta-analysis [6,7,10,11,12,13,14,15,16,20,21].

**Table 1 biomedicines-11-01178-t001:** Characteristics of included studies and participants.

Reference	N	Age, Mean ± SD	Male Gender (%)	Nationality	AAA Diameter (mm), Mean ± SD	Hypertension (%)	Diabetes (%)	Ever Smoked (%)	Current Smoker (%)	Statins (%)	Aspirin (%)	sBP, Mean ± SD	dBP, Mean ± SD	LDL-c, Mean ± SD
Nicolajsen, 2022 [18]	33,393	73.6 @	25,353 (75.9)	Danish	NR	27.3	7.7	NR	NR	54.8 #	60.8 ^#	NR	NR	NR
Nastasi, 2021 [6]	583	73.5 (67.8–78.3) *	476 (82.0)	Australian	42 (36.0–48.0) *	77.4	24.2	85.2	NR	70.5	66.6 ^	NR	NR	2.4 (1.8–3.0) *
Ko, 2021 [17]	10,822	63.9 ± 11.7	6925 (64.0)	Korean	NR	73.8	18.1	49.4	27.1	NR	NR	127.89 ± 16.6	78.3 ± 10.9	3.0 ± 1.3
Oliver-Williams, 2019 [10]	18,652	66.8 °	18,652 (100.0)	British	NR	NR	NR	89.8	35	62.4	39.5	142.3 °	83.0 °	NR
Elbasty, 2019 [11]	101	85–89 $	88 (87.1)	British	45 ± 9.2	55.5	14.9	48.5	NR	85.1	51.5	NR	NR	NR
Takigawa, 2012 [19]	157	72.7 ± 7.5	128 (82.0)	Japanese	50.1 ± 8.8	77.0	11.0	77.0	NR	31.0	NR	137.5 ± 16.2	75.6 ± 10.4	3.4 ± 0.9
Parr, 2011 [7]	98	73 (67–77) *	75 (76.5)	Australian	47.2 (34.9–54.6) *	76.5	20.4	87.8	NR	66.3	63.3	NR	NR	NR
Forsdahl, 2010 [12]	345	NR	NR	Norwegian	NR	NR	NR	NR	NR	NR	NR	NR	NR	NR
Brown, 2008 ! [13]	527	69.2 ± 4.4	434 (82.0)	British	46.1 ± 3.7	40.0	3.0	94.0	34.0	NR	29.0	156.7 ± 26.6	NR	NR
Baumgartner, 2008 [14]	1722	73.4 ±7.8	1389 (80.7)	Multinational ∍	NR	83.5	29.4	81.0	20.9	71.2	65.2	135.9 ± 19.0	76.2 ± 11.2	NR
Lindholt, 2007 [20]	191	65.3–70.7 *	191 (100.0)	Danish	31–41 *	NR	NR	NR	NR	NR	NR	NR	NR	NR
Newman, 2001 [15]	416	75 (65–93) *	252 (61.0)	American	NR	45.0	10.0	NR	14.7	NR	NR	NR	NR	NR
Brady, 2001 [21]	2305	NR	NR	British	NR	NR	NR	NR	NR	NR	NR	NR	NR	NR
Galland, 1998 [16]	267	NR	NR	British	NR	NR	NR	NR	NR	NR	NR	NR	NR	NR
Weighted average	-	70.1	81.0	-	43.9	41.1	11.1	75.7	27.4	58.1	53.5	136.9	80.9	3.0

N—number of participants; AAA—abdominal aortic aneurysm; sBP—systolic blood pressure; dBP—diastolic blood pressure; LDL-c—low density lipoprotein cholesterol. Age reported as mean ± standard deviation/median (interquartile range). @—age reported as weighted average for median age of four separate cohorts across different time periods. *—reported as median (interquartile range). $—Reported range without median. °—standard deviation not reported. ∍—Multinational—including patients from United States of America and Europe. #—Medication prescription was reported for 33,296 out of the total 33,393 patients in the study. ^—reported as antiplatelet medication, not specifically aspirin. !—Patient cohort characteristics data sourced from United Kingdom Small Aneurysm Trial [22]. NR—Not reported.

**Table 2 biomedicines-11-01178-t002:** Risk of bias of included studies.

Reference	Objective Defined	Study Design	Sample Size Estimation	Unrepaired AAA	MACE Reporting	Primary Outcome	Participant Characteristics	Follow Up	Statistical Methods	Total	Percentage	Risk of Bias
Nicolajsen, 2022 [18]	2	1	0	0	1	0	2	1	2	9	50.0%	Moderate
Nastasi, 2021 [6]	2	2	0	2	2	2	2	2	2	16	88.9%	Low
Ko, 2021 [17]	2	1	0	0	1	1	2	NR	2	9	50.0%	Moderate
Oliver-Williams, 2019 [10]	2	2	0	2	0	2	1	2	1	12	66.7%	Moderate
Elbasty, 2019 [11]	2	2	0	2	1	2	2	2	0	13	72.2%	Low
Takigawa, 2012 [19]	2	1	0	0	1	1	2	NR	2	9	50.0%	Moderate
Parr, 2011 [7]	2	2	0	2	2	2	2	1	2	15	83.3%	Low
Forsdahl, 2010 [12]	2	2	0	2	0	2	0	2	2	12	66.7%	Moderate
Brown, 2008 [13]	2	2	0	2	2	2	0	2	1	13	72.2%	Low
Baumgartner, 2008 [14]	2	2	0	0	2	2	2	1	1	12	66.7%	Moderate
Lindholt, 2007 [20]	2	1	0	0	0	1	0	2	1	7	38.9%	High
Newman, 2001 [15]	2	2	0	2	0	2	2	2	1	13	72.2%	Low
Brady, 2001 [21]	2	1	0	2	2	2	0	1	2	12	66.7%	Moderate
Galland, 1998 [16]	2	2	0	2	2	2	0	2	0	12	66.7%	Moderate

AAA—abdominal aortic aneurysm; MACE—major adverse cardiovascular events; NR—not reported.

**Table 3 biomedicines-11-01178-t003:** Number of cardiovascular events and deaths of participants with unrepaired abdominal aortic aneurysm.

Study	N	Duration of Follow Up (years)	MI, n (%)	Stroke, n (%)	CV Death, n (%)	AAA Rupture, n (%)	Surgery Related Death, n (%)	Total AAA Death, n (%)	All Cause Mortality, n (%)
Nicolajsen, 2022 [18]	33,393	2.0	864 (2.6)	796 (2.4)	NR	NR	NR	NR	8404 (25.2)
Nastasi, 2021 [6]	583	4.1	47 (8.1)	23 (3.9)	52 (8.9)	NR	NR	NR	NR
Ko, 2021 ! [17]	10,822	NR	NR	NR	793 (7.3)	NR	NR	NR	1677 (15.5)
Oliver-Williams, 2019 [10]	18,652	2.5	NR	NR	258 (1.4)	31 (0.2)	NR	29 (0.2)	980 (5.3)
Elbasty, 2019 [11]	101	4.7	NR	NR	8 (7.9)	2 (2.0)	1 (1.0)	3 (3.0)	26 (25.7)
Takigawa, 2012 * [19]	157	NR	46 (29.3)	40 (25.5)	NR	NR	NR	NR	NR
Parr, 2011 [7]	98	3.0	13 (13.3)	1 (1.0)	9 (9.2)	NR	NR	NR	NR
Forsdahl, 2010 [12]	345	10.1	NR	NR	76 (22.0)	NR	NR	20 (5.5)	130 (36.0)
Brown, 2008 @ [13]	527	12.0	58 (11.0)	27 (5.1)	155 (29.4)	23 (4.4)	30 (5.7)	53 (10.1)	352 (66.8)
Baumgartner, 2008 # [14]	1629 *	1.0	23 (1.4)	25 (1.5)	34 (2.1)	NR	NR	NR	NR
Lindholt, 2007 [20]	191	5.2	NR	NR	39 (20.4)	NR	NR	7 (3.7)	49 (25.7)
Newman, 2001 [15]	416	4.5	NR	NR	58 (13.9)	6 (1.4)	NR	6 (1.4)	110 (26.4)
Brady, 2001 [21]	2305	1.7	NR	NR	175 (7.6)	40 (1.7)	84 (3.6)	40 (1.7)	259 (11.2)
Galland, 1998 [16]	267	5.0	NR	NR	36 (13.5)	1 (0.4)	2 (0.8)	3 (1.1)	NR

N—number of participants. MI—myocardial infarction. CV—cardiovascular. AAA—abdominal aortic aneurysm. NR—not reported. AAA rupture—death due to AAA rupture. Surgery-related death—death within 30 days after AAA repair. Total AAA death—death due to AAA rupture or surgery-related death. All results are reported as: total number of events (percentage %). * of the total 1722 patients in the study, outcomes were only reported for 1629. ! Study reported fatal MI and fatal stroke, so to calculate cardiovascular death, added fatal MI and fatal stroke (Fatal MI = 350, fatal stroke = 443). @ 28 of the 53 AAA-related deaths were repair related. # results reported as percentages and converted to whole numbers by multiplying sample size by percentage and rounding to the nearest integer.

**Table 4 biomedicines-11-01178-t004:** Leave one out analysis of the incidence of cardiovascular death in patients with AAA.

Study Omitted	Incidence Rate (95% CI)	Cook.d
Omitting Nastasi, 2021 [6]	2.32 (1.58–3.40)	0.001
Omitting Oliver-Williams, 2019 [10]	2.74 (2.27–3.32)	0.958
Omitting Elbasty, 2019 [11]	2.36 (1.63–3.43)	0.019
Omitting Parr, 2011 [7]	2.25 (1.55–3.27)	0.016
Omitting Forsdahl, 2010 [12]	2.32 (1.58–3.41)	0.001
Omitting Brown, 2008 [13]	2.29 (1.56–3.67)	0.001
Omitting Baumgartner, 2008 [14]	2.33 (1.59–3.41)	0.003
Omitting Lindholt, 2007 [20]	2.18 (1.52–3.15)	0.093
Omitting Newman, 2001 [15]	2.24 (1.53–3.27)	0.030
Omitting Brady, 2001 [21]	2.14 (1.50–3.06)	0.167
Omitting Galland, 1998 [16]	2.27 (1.55–3.32)	0.008

Incidence rate—reported as per 100 person-years. 95% CI—95% confidence interval. Cook.d—Cook’s distance: measures the change in the estimated regression coefficients when a particular observation is removed from the dataset, a value greater than 0.50 was considered influential.

**Table 5 biomedicines-11-01178-t005:** Metaregression for MACE and associated risk factors.

Metaregression		Intercept	SE	*p* Value
**Cardiovascular death**	Diabetes	−3.61	0.01	0.35
	HTN	−3.46	0.004	0.27
	Aspirin	−4.83	0.02	0.77
	Statin	−5.74	0.05	0.62
	Year	−95.5	0.02	0.01
**Myocardial infarction**	Diabetes	−5.16	0.02	0.23
	HTN	−5.05	0.01	0.66
	Aspirin	−6.27	0.01	<0.01
	Year	−14.53	0.05	0.91
**Stroke**	Diabetes	−4.54	0.02	0.27
	HTN	−4.88	0.01	0.19
	Aspirin	−3.12	0.02	0.17
	Year	−58.95	0.04	0.55

Diabetes—diagnosis of diabetes; HTN—diagnosis of hypertension; Aspirin—aspirin use; Statin—statin use; SE—standard error.

## Data Availability

All supporting data are available within the manuscript and Appendix A.

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
