# Peer review of "A Systematic Review and Meta-Analysis of the Incidence and Risk Factors for Major Adverse Cardiovascular Events in Patients with Unrepaired Abdominal Aortic Aneurysms"

_biomedicines, 2023, doi:10.3390/biomedicines11041178_

Round 1

Reviewer 1 Report

The Authors proposed an interesting paper, in which they performed a systematic review and meta-analysis evaluating the incidence and the risk factors for MACE occurrence in patients with unrepaired AAA.

The topic is of interest and the results confirm that patients with AAA deserve (even if they do not reach the AAA diameter threshold for treatment) a specific attention not only for the AAA-related complications but also for the generale cardiovascular risk. 

The paper is well constrocted and the methodology is clear and well reported. 

I red the article with great interest and  I have no major concers about it. 

Reviewer 2 Report

The proposed manuscript is devoted to a review evaluating the incidence rate of major adverse cardiovascular events in patients with unrepaired abdominal aortic aneurysm. The authors investigate possible correlations of these major adverse cardiovascular events with modifiable risk factors and long term medication use.

Preliminaries to the research area are provided. Information related to the abdominal aortic aneurysm, main factors of its occurrence, methods of assessment of the corresponding rupture risk and growth is provided. The role of the screening programs for identification of abdominal aortic aneurysm is discussed. The relationships between abdominal aortic aneurysm and other diseases  such as coronary heart and cerebrovascular diseases are reviewed providing results of previous studies.

The methodology of the proposed study is described in detail, in particular the literature search strategy, the extraction of data and outcomes, the quality assessment and the analysis of data.

The results of the statistical analysis and their characteristics are presented. A detailed discussion of the results is given and corresponding conclusions are formulated.

The presentation of the main results is clear and comprehensive. From a formal point of view, all the contents seems to be correct. The results are valuable and worthy of being published taking into account their possible applications in clinical practice and health care, in particular for secondary cardiovascular prevention in patients with unrepaired abdominal aortic aneurysm.

Minor revisions are suggested to improve the quality of the exposition:

p. 1, lines 6-7: I suggest the degrees of the authors to be removed from their list.

p. 4-5: Under Table 1 there are caption rows which seem to be related to another Table, it should be verified.

p. 14-16: The formatting of the references is different (e.g. in some of them the full names of the journals are given but in other – only of their abbreviations), it should be unified in accordance with the requirements of the Journal.

p. 14-16: The numbering of the last pages is wrong.

­

Reviewer 3 Report

I consider the article submitted for review to be well-prepared.
The topic of MACE among unrepaired abdominal aortic aneurysms taken up by the authors is a significant issue.

Weakness of this review article:

1. The title needs a slight correction: "A systematic review and meta-analysis of the incidence of and risk factors for major adverse cardiovascular events in patients with unrepaired abdominal aortic aneurysms" - "incidence of and risk factors" (?) - it sounds strange.

2. Abstract is quite "bland." It starts with "This review assessed" and ends with "This review suggests" - so what exactly did the authors determine? The authors conclude that "more effective methods are needed in secondary cardiovascular prevention in this population." More effective than?

A bit disappointing is the final result of searching the literature - because the authors summarize the research with an unequivocal statement that "incidence of MACE is significantly increased in patients with unrepaired AAA and more effective methods are needed in secondary cardiovascular prevention in this population."
Some deeper reflection is needed, maybe their commentary. Otherwise, although it is a very well-methodologically prepared study, it does not bring much.

3. I didn't notice any "risk factors" in the conclusions. Have the authors identified these risk factors?
I was interested in this passage of text, "The meta-regression found a statistically significant inverse correlation between aspirin use and stroke (Supplementary Table 2). This contradicts the current consensus in the literature, which reports a positive correlation between aspirin use and stroke... Small numbers likely confound this finding, the lack of information on the effectiveness of the drugs in controlling modifiable risk factors, drug adherence, and dose." -  I have to admit that this is not very educative - it does not help the reader to understand that even there exists a positive correlation between stroke and aspirin use, the meta-regression showed the opposite. And then "explanation on that" that probably this is because of a little group for analyzing. I don't think this should be in the article's main body because it's confusing.
4. I did not find Supplementary Table 2 in the attachments - although the authors refer to it in the text.

5. Figure 3 is very illegible.

Summarizing, the review article should be, in my opinion, more profound and should give a better explanation of the underlying processes. This article does not explain and does not contribute much.

Round 2

Reviewer 3 Report

The article has been corrected according to my comments.
I'm satisfied with the changes. I have no more comments. I recommend accepting the article as it stands.